# Barriers to integration of bioinformatics into undergraduate life sciences education: A national study of US life sciences faculty uncover significant barriers to integrating bioinformatics into undergraduate instruction

Jason J. Williams[1], Jennifer C. Drew[2‡], Sebastian Galindo-Gonzalez[3‡], Srebrenka Robic[4‡], Elizabeth Dinsdale[5‡], William R. Morgan[6‡], Eric W. Triplett[2‡], James M. Burnette III[7‡], Samuel S. Donovan[8‡], Edison R. Fowlks[9‡], Anya L. Goodman[10‡], Nealy F. Grandgenett[11‡], Carlos C. Goller[12‡], Charles Hauser[13‡], John R. Jungck[14‡], Jeffrey D. Newman[15‡], William R. Pearson[16‡], Elizabeth F. Ryder[17‡], Michael Sierk[18‡], Todd M. Smith[19‡], Rafael Tosado-Acevedo[20¤a‡], William Tapprich[21‡], Tammy C. Tobin[22‡], Arlín Toro-Martínez[23‡], Lonnie R. Welch[24‡], Melissa A. Wilson[25‡], David Ebenbach[26‡], Mindy McWilliams[26‡], Anne G. Rosenwald[27], Mark A. Pauley[28¤b]*

1 Cold Spring Harbor Laboratory, Cold Spring Harbor, NY, United States of America, 2 Microbiology and Cell Science Department, University of Florida, Gainesville, FL, United States of America, 3 Department of Agricultural Education and Communication, University of Florida, Gainesville, FL, United States of America, 4 Department of Biology, Agnes Scott College, Decatur, GA, United States of America, 5 Department of Biology, San Diego State University, San Diego, CA, United States of America, 6 Department of Biology, College of Wooster, Wooster, OH, United States of America, 7 University of California, Riverside, Riverside, CA, United States of America, 8 Department of Biological Sciences, University of Pittsburgh, Pittsburgh, PA, United States of America, 9 Department of Biological Sciences, Hampton University, Hampton, VA, United States of America, 10 Department of Chemistry and Biochemistry, California Polytechnic State University, San Luis Obispo, CA, United States of America, 11 Department of Teacher Education, University of Nebraska at Omaha, Omaha, NE, United States of America, 12 Department of Biological Sciences, North Carolina State University, Raleigh, NC, United States of America, 13 Department of Biological Sciences, Bioinformatics Program, St. Edward's University, Austin, TX, United States of America, 14 Departments of Biological Sciences and Mathematical Sciences, University of Delaware, Newark, DE, United States of America, 15 Department of Biology, Lycoming College, Williamsport, PA, United States of America, 16 Department of Biochemistry and Molecular Genetics, University of Virginia School of Medicine, Charlottesville, VA, United States of America, 17 Biology and Biotechnology Department, Worcester Polytechnic Institute, Worcester, MA, United States of America, 18 Bioinformatics Program, Saint Vincent College, Latrobe, PA, United States of America, 19 Digital World Biology, PMB, Seattle, WA, United States of America, 20 Department of Natural Sciences, Inter American University of Puerto Rico, Metropolitan Campus, San Juan, PR, United States of America, 21 Department of Biology, University of Nebraska at Omaha, Omaha, NE, United States of America, 22 Department of Biology, Susquehanna University, Selinsgrove, PA, United States of America, 23 Department of Biology, Chemistry, and Environmental Sciences, Inter American University of Puerto Rico, San Germán Campus, San Germán, PR, United States of America, 24 Department of Computer Science, Ohio University, Athens, OH, United States of America, 25 School of Life Sciences, Arizona State University, Tempe, AZ, United States of America, 26 Center for New Designs in Learning and Scholarship, Georgetown University, Washington, DC, United States of America, 27 Department of Biology, Georgetown University, Washington, DC, United States of America, 28 School of Interdisciplinary Informatics, University of Nebraska at Omaha, Omaha, NE, United States of America

☯ These authors contributed equally to this work.
¤a Current address: National Center for Emerging and Zoonotic Infectious Diseases, Centers for Disease Control and Prevention, San Juan, PR, United States of America.
¤b Current address: Division of Undergraduate Education, Directorate for Education and Human Resources, National Science Foundation, Alexandria, VA, United States of America.
‡ These authors also contributed equally to this work.
* mpauley@nfs.gov



**Data Availability Statement:** Data are available on the NIBLSE resposity on GitHub, https://github.com/niblse.

**Funding:** This material is based upon work supported by the National Science Foundation under Grant no. 1539900 to E.D., M.W., A.G.R., E.W.T., and W.T. The funders had no role in study design, data collection and analysis, decision to publish, or preparation of the manuscript. A commercial company, Digital World Biology, provided support in the form of salary for author TMS but did not have any additional role in the study design, data collection, and analysis, decision to publish, or preparation of the manuscript. The specific roles of this author are articulated in the "author contributions" section.

**Competing interests:** We declare that author TMS has an affiliation with a private company, Digital World Biology (DWB). As noted in the Funding Statement, DWB provided support for this work in the form of salary for TMS. This affiliation does not alter our adherence to PLoS ONE policies on sharing data and materials.

# Abstract

Bioinformatics, a discipline that combines aspects of biology, statistics, mathematics, and computer science, is becoming increasingly important for biological research. However, bioinformatics instruction is not yet generally integrated into undergraduate life sciences curricula. To understand why we studied how bioinformatics is being included in biology education in the US by conducting a nationwide survey of faculty at two- and four-year institutions. The survey asked several open-ended questions that probed barriers to integration, the answers to which were analyzed using a mixed-methods approach. The barrier most frequently reported by the 1,260 respondents was lack of faculty expertise/training, but other deterrents—lack of student interest, overly-full curricula, and lack of student preparation—were also common. Interestingly, the barriers faculty face depended strongly on whether they are members of an underrepresented group and on the Carnegie Classification of their home institution. We were surprised to discover that the cohort of faculty who were awarded their terminal degree most recently reported the most preparation in bioinformatics but teach it at the lowest rate.

## Introduction

Bioinformatics, an interdisciplinary field that combines aspects of biology, statistics, mathematics, and computer science, is becoming increasingly important for research efforts in all areas of biology [1,2]. Biology students graduating with bioinformatics experience have more employment opportunities available to them [3] and are better prepared for graduate studies in life sciences fields. It has also been suggested that students graduating with degrees in molecular biology and biochemistry should have some familiarity with bioinformatics [4]. With the growing emphasis on "big data" in biology, there is more demand for researchers in the life sciences with training in bioinformatics. However, many life sciences students earn their degrees with little exposure to it [5–7].

The Network for Integrating Bioinformatics into Life Sciences Education (NIBLSE, "nibbles"; https://niblse.org), a National Science Foundation Research Coordination Network, is a group of US education and private sector professionals in biology, bioinformatics, and computer science dedicated to making bioinformatics an integral component of instruction in the life sciences nationwide. Our approach involves developing instructional strategies for undergraduates to gain experience in bioinformatics, working to address barriers to the implementation of those strategies, and designing assessment instruments to evaluate the impact on student preparation [8].

In the US, bioinformatics instruction has predominately been provided at the graduate level [9–11]. Although we are aware that undergraduate bioinformatics courses are becoming more common, there has been little effort to integrate this interdisciplinary field broadly into undergraduate biology curricula. To further this integration, a better understanding of the barriers preventing its inclusion is necessary. We thus surveyed life sciences faculty at two- and four-year institutions across the US. Part of the survey consisted of open-ended, free-response questions that probed barriers to the integration of bioinformatics. Individual answers to these questions were qualitatively analyzed for specific barriers that deductively arose from the overall set of responses. (Example responses are provided in S1 Responses) The number of answers

that were judged to refer to these key concepts were counted, and the counts were analyzed with respect to other data collected in the survey (see Materials and Methods). Given the number of valid responses to the survey—1,231; 1% to 2% of all US biological sciences faculty [12]—our findings provide a national consensus view. Below we discuss the major barriers uncovered and then describe efforts we and others are taking to address them.

## Results

NIBLSE was founded on the premise that bioinformatics is and will continue to be essential for undergraduate biology education. One of the first questions in the survey asked whether respondents shared this view. Approximately 95% of survey respondents (Fig 1) agreed with the statement "Bioinformatics should be integrated into undergraduate life sciences education." At the same time, however, only a third, 32%, said that they currently teach courses with at least some bioinformatics content.

The survey included four open-ended, free-response questions that asked faculty about the barriers they face in including bioinformatics in their teaching (Table 1). As described in Materials and Methods, the responses to these questions were analyzed qualitatively for specific barriers (e.g., "Lack of expertise/training" and "Lack of time") that arose deductively from the overall set of responses. The categories Question 1 generated are given in Table 2. The categories were then combined into super-categories. Responses generated eight super-categories: "Faculty Issues," "Student Issues," "Curriculum Issues," "Facilities Issues," "Resource Issues," "Institutional Issues," "State Issues," and "Accreditation Issues." The number of responses that

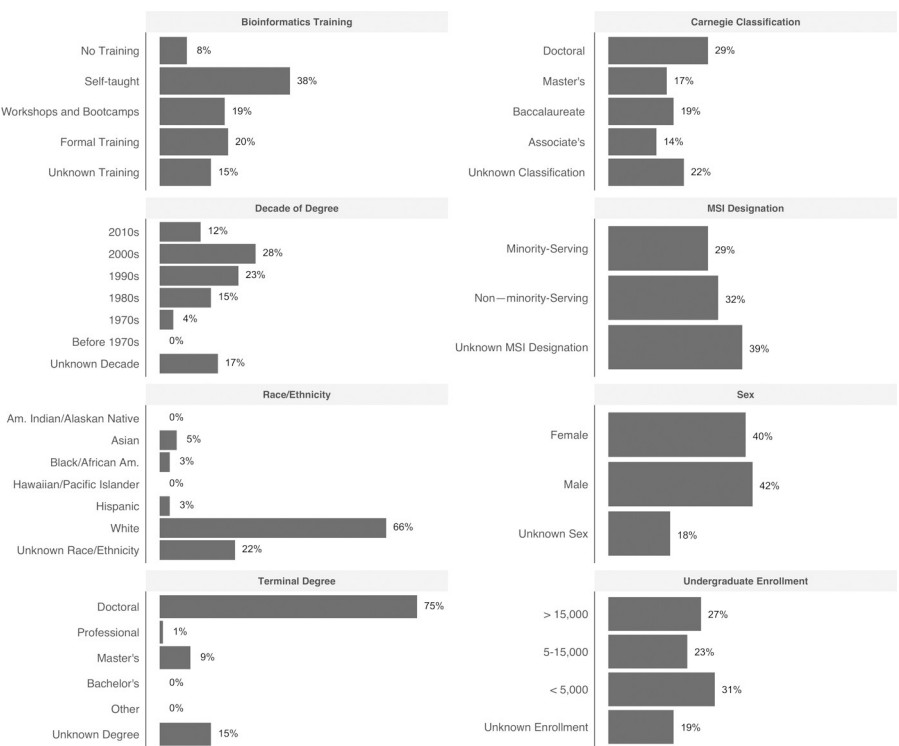

**Fig 1. Summary demographics.** Summary demographics shown as percentages of respondents (*n* = 1,231, the total number of US respondents). The composite survey respondent is a white male or female PhD, self-taught in bioinformatics, with their degree earned in 2000–2009. S/he works at a non-minority-serving, doctoral-granting institution with an undergraduate enrollment of less than 5,000.

**Table 1. Survey questions about barriers faculty face in integrating bioinformatics into undergraduate life sciences instruction.**

| Question | Number of Responses |
|---|---|
| 1. In your opinion, what do you think are the most important challenges currently facing those educating undergraduate life scientists in bioinformatics? | 734 (59.6%) |
| 2. Please describe briefly [your opinion about the need for additional undergraduate courses with bioinformatics content at your institution]; include any barriers to development and/or implementation. | 364 (29.6%) |
| 3. What is preventing you from including bioinformatics content in these courses? | 313 (25.4%) |
| 4. At your current institution, do you face any technical barriers in teaching bioinformatics, e.g., availability of a computer lab, different operating systems, access to high performance computing for teaching, IT support? Please describe. | 511 (41.5%) |

Question 3 was only asked if a respondent indicated they were not currently integrating bioinformatics into their courses. The responses to these questions were analyzed qualitatively for specific barriers.

mentioned a given category of barrier was then counted. Although not every respondent answered all the open-ended questions and some didn't answer any, there were almost 2,000 responses to the four questions (Table 3). Here, we describe our findings with respect to the two sets of barriers, "Faculty Issues" and "Student Issues," that came up the most frequently, then describe others that were also commonly reported.

**Table 2. Super-categories and categories of barriers in responses to Question 1.**

| *Question 1: In your opinion, what do you think are the most important challenges currently facing those educating undergraduate life scientists in bioinformatics?* | |
|---|---|
| **Super-category** | **Category** |
| Faculty issues | Unspecified<br>No expertise/training<br>Time<br>Differences of opinion<br>Content development<br>Not enough faculty |
| Facilities issues | Unspecified<br>Computer labs limited or not available<br>Computers are too old/inadequate |
| Resource issues | Unspecified<br>Access to appropriate software<br>Funding (general)<br>Funding (software license fees) |
| Student issues | Lack of appropriate background knowledge/skills<br>No interest in bioinformatics<br>Intimidated by topic<br>Multitude of varying student backgrounds<br>Lack of basic computing knowledge<br>Career prospects |
| Curriculum issues | Unspecified<br>Difficulties in communication of computational processes in biology<br>Too much content in life science curriculum<br>How quickly the material changes/how quickly the technology changes<br>Access to developed bioinformatics lesson plans/bioinformatics curriculum<br>Making computer science courses consistently relevant<br>Too much curriculum influence from professional schools |
| Institutional/Departmental Support issues | Unspecified<br>Interdepartmental cooperation<br>No IT support |

**Table 3. Categories by question.**

| Free-response question | Q1. Educator challenges | Q2. Barriers to implementation | Q3. Barriers to inclusion* | Q4. Technical barriers |
|---|---|---|---|---|
| **Total number free-text comments** (percentage of respondents writing comments; n = 1,231) | **734** (59.6) | **364** (29.6) | **313** (25.4) | **511** (41.5) |
| **Non-responders** (Percentage of survey participants not completing this question; n = 1,231) | **497** (40.3) | **867** (70.4) | **918** (74.6) | **720** (58.5) |
| **Number of respondents identifying with a Barrier Category** (percentage of unique respondents in each category; n = 1,231) | | | | |
| Faculty Issues | **358** (29.1) | **222** (18) | **308** (25) | **36** (2.9) |
| Student Issues | **295** (24) | **62** (5.0) | **69** (5.6) | **19** (1.5) |
| Curriculum Issues | **227** (18.4) | **118** (9.6) | **226** (18.4) | **8** (0.7) |
| Resource Issues | **77** (6.3) | **36** (2.9) | **84** (6.8) | **139** (11.3) |
| Facilities Issues | **53** (4.3) | **22** (1.8) | **18** (1.5) | **186** (15.1) |
| Institutional Issues | **27** (2.2) | **43** (3.5) | **3** (0.2) | **133** (10.8) |
| State Issues | **0** (0) | **0** (0) | **2** (0.2) | **0** (0) |
| Accreditation Issues | **0** (0) | **0** (0) | **1** (0.1) | **0** (0) |

*Question 3 was only shown to n = 591 respondents who indicated they were not integrating bioinformatics into their teaching.

Respondents answered up to four free-response questions about the barriers they face in integrating bioinformatics into their instruction. For a given question, we report the total number of free-text comments and overall response rate. When tallying responses, a single respondent's answer may have been coded into multiple super-categories—multiple barriers could be reported in a single response—but for any one of the eight (see narrative), an individual response appears no more than once. The percentage of responses reporting a given category is shown as a percentage of the total number of valid survey responses (n = 1,231). The numbers are likely undercounts since non-entries, including those from respondents who did not complete the survey, were taken to mean that the respondent did not experience a barrier (see Materials and Methods).

As shown in Figs 2 and 3, items in the super-category faculty issues were the most commonly reported barriers faculty face. This was true whether the respondent data were stratified by sex, race, ethnicity, institutional Carnegie Classification (institution type), minority-serving institution status, size of the undergraduate population, or geographic region (Fig 1). Under faculty issues, "Lack of expertise/training" was by far the most common barrier at all institution types except for doctoral-granting institutions; at doctoral institutions, one of the student issues, "Lack of skills/knowledge" was the most frequently reported (Fig 4).

We hypothesized that faculty who had earned their terminal degree most recently would report the highest amount of formal training in bioinformatics. Nearly 50% of faculty who earned their highest degree in 2010–2016 reported some kind of formal training (undergraduate or graduate courses and/or certificates), compared to 35% of the 2000–2009 cohort and decreasing thereafter (Table 4) (n = 968). Despite this level of formal training, faculty who earned their degrees most recently were the least likely (P = 0.003) (n = 908) to report teaching dedicated bioinformatics courses or teaching courses with some bioinformatics content (Fig 5). This is the case even though faculty from the 2010–2016 cohort teach at all types of institutions at about the same percentages (Table 5).

When we looked closely at who is integrating bioinformatics into their teaching—either teaching a dedicated course or incorporation into other courses—those who described themselves as self-taught are the most likely group to integrate at just over 18%. Thirteen percent of those with workshop or bootcamp training reported integration, and only 11% of respondents with formal training integrate bioinformatics into their teaching. Only a single individual with no training reported any form of integration (n = 877).

With respect to sex, females and males (n = 842) reported integrating bioinformatics at similar rates (20% female, 23% male). Females are more likely to be teaching at associate's

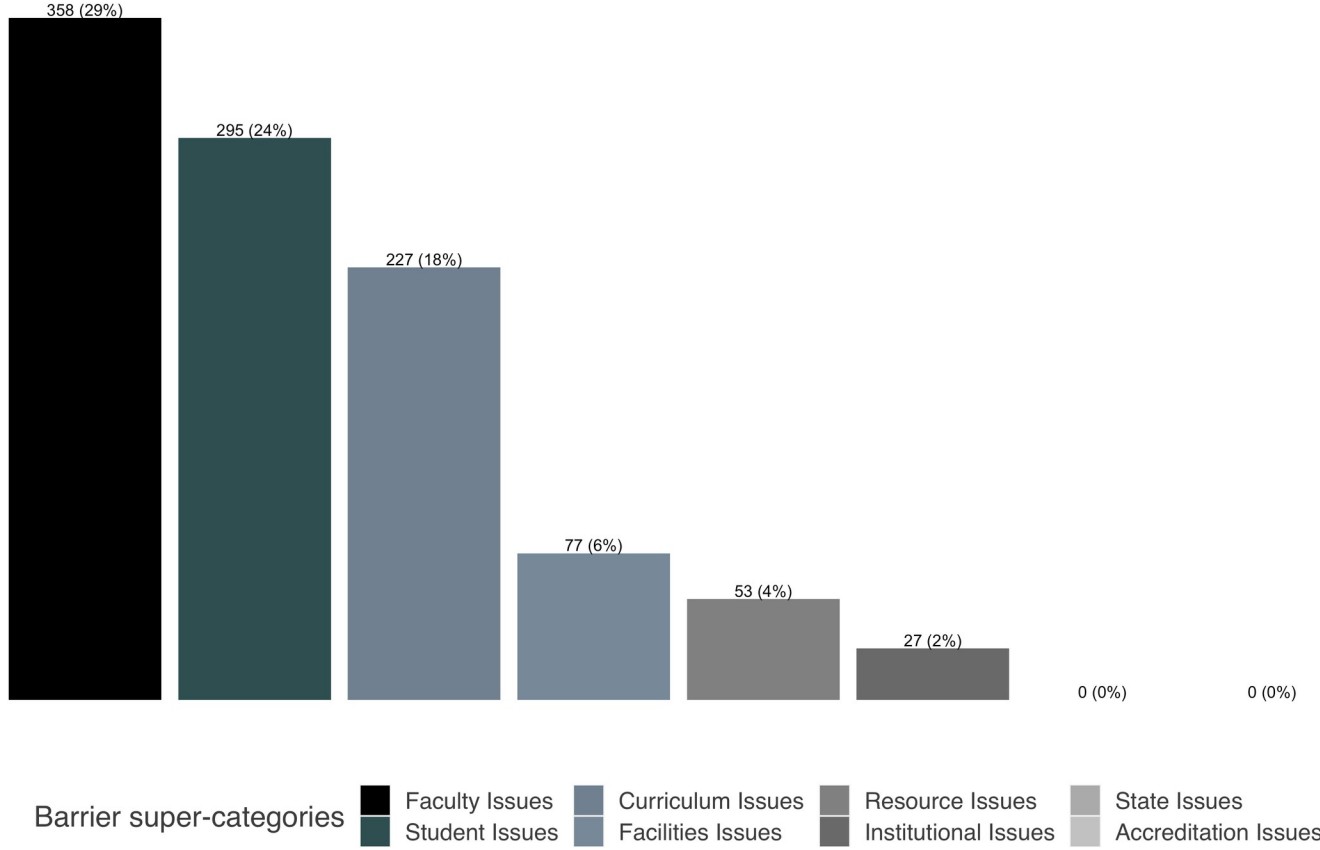

**Fig 2. Summary of most commonly reported barriers by super-category.** The number and percentage (in brackets) of respondents with comments corresponding to one of eight barrier super-categories are shown for Question 1. Seven hundred thirty-four respondents (of a total $n = 1,231$) provided a free-text response for this question. As shown, faculty-related barriers were the barriers reported most frequently.

institutions (12% female vs. 7% male) and less likely to be teaching at doctoral-granting institutions (15% female vs. 22% male) ($n = 929$). The number of females obtaining terminal degrees has increased—7% of respondents who reported earning their terminal degree in the 1980s were female compared to 20% who graduated in the 2000s—with the latest cohort (2010–2016) having nearly equal numbers of males (7%) and females (9%) ($n = 929$). Females did not report training as a barrier significantly more than males did (30% vs. 26%) ($n = 1013$) but reported lack of access to computer labs at double the percentage of males (Question 4, Table 1; Fig 6). Slightly fewer females than males reported being self-taught in bioinformatics (20% female vs. 25% male), but both sexes are nearly evenly split in the other forms for training (workshops—12% female, 10% male; formal training—11% female, 12% male) or no training (5% female, 4% male) ($n = 1013$).

To determine if the barriers faculty face depend on whether they are members of an underrepresented minority (URM) in science, technology, engineering, and mathematics (STEM), we compared the responses of URM to non-URM faculty. (For this study, we considered the following groups to be underrepresented in STEM: Blacks, Hispanics, American Indians and Alaska Natives, and Native Hawaiians and other Pacific Islanders [13–15].) Because the number of respondents identifying as URMs was small—less than 7% of the total, a result that mirrors the lack of diversity in US life sciences faculty reported elsewhere [16]—we combined these respondents into a single group for analysis. We found that URM faculty reported

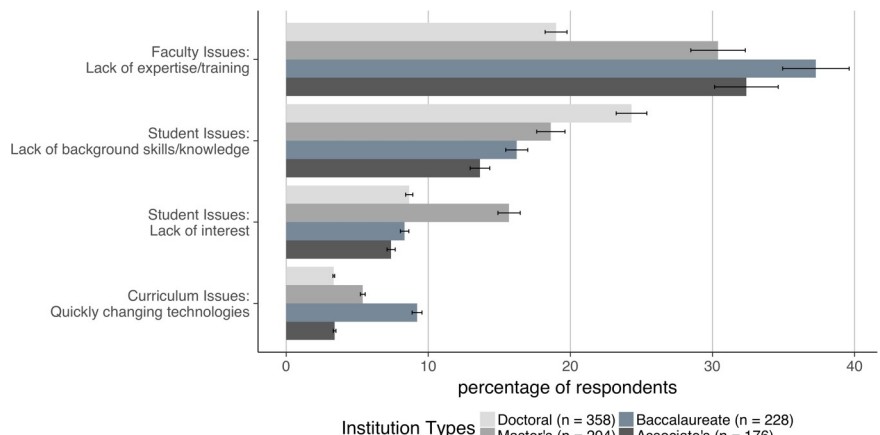

**Fig 3. Barriers reported across questions.** Faculty-related barriers were consistently the top reported barriers in all questions, except Question 4, which asked specifically about technical barriers. *Question 3 was only shown to respondents who indicated they were not currently integrating bioinformatics into their teaching.

**Fig 4. The biggest barrier at most institution types is lack of expertise/training.** The figure shows four barriers that faculty at the different institution types experience the most differently. The margin of error, as the interval estimate of population proportion, was calculated at the 95% confidence level and is represented as error bars. Of the four, the lack of training/expertise was by far the most common problem at all institution types except for doctoral-granting institutions, where students' lack of background skills/knowledge was the most common. Also of note is that students at master's institutions seem less interested in bioinformatics than those at other institution types. See the Discussion for our thoughts on these two issues.

**Table 4. Characteristics of faculty cohorts stratified by degree year.**

| Decade of Highest Degree Earned | Formal Bioinformatics Training (%) | Faculty Integrating Bioinformatics (%) |
|---|---|---|
| 1980–1989 | 8.4 | 35.4 |
| 1990–1999 | 11.3 | 41.9 |
| 2000–2009 | 35.1 | 41.7 |
| 2010–2016 | 48.3 | 25.2 |

As shown in Fig 1, some faculty respondents earned their terminal degree before 1980, but the number was small, so that cohort is not included here.

training as a barrier much more frequently than non-URMs—42% vs. 28% ($n = 961$), respectively. Comparing faculty at minority-serving institutions (MSIs) with those at non-MSIs, MSI faculty report faculty issues as a barrier at a slightly lower rate than faculty at non-MSIs.

Faculty described several ways in which time was a barrier, including lack of instructional time to teach more material, lack of time for additional training, and lack of time for course development or restructuring. These responses were captured in the category "Lack of time," a subcategory of faculty issues (Fig 2 and Table 2).

The student issues super-category was the second most frequently mentioned set of barriers after faculty issues (Fig 2). Two particular issues were commonly reported: students' lack of background skills and knowledge, mentioned most frequently by faculty at doctoral-granting institutions, and students' lack of interest, mentioned most frequently by faculty at master's institutions (Fig 4). When we delved more deeply into the individual responses, we found that faculty at different institution types had different concerns, likely reflecting different expectations of their students. For example, faculty at doctoral-granting institutions were most concerned about their students' lack of statistics knowledge and programming skills, whereas those at associate's colleges mentioned their students' lack of basic mathematics skills most often. In addition, we found that faculty teaching a dedicated bioinformatics course reported that their students lack the appropriate background at a much higher rate than those not teaching a dedicated course (Fig 7).

Many respondents reported barriers we grouped under the super-category curriculum issues (Fig 2). The two most frequently mentioned issues were "Communication difficulties," specifically differences in the way biologists and computer scientists approach problems and communicate, and "Too much content," referring to the difficulties inherent in including additional material in existing courses. Many respondents also mentioned "Quickly changing technologies," alluding to the difficulties in keeping up with this rapidly changing field both in terms of training and access to software. This barrier was especially problematic at baccalaureate colleges (Fig 4), where faculty often have higher teaching loads across a wider range of subjects and fewer resources than those at research institutions. Interestingly, this barrier seemed to be less of a problem at associate's-granting colleges, possibly reflecting the prescribed curriculum found at many two-year schools. Finally, respondents also mentioned "Institutional support issues," including fellow faculty who do not feel that bioinformatics has a place in life sciences curricula and lack of support from administrators for resources such as training for faculty or hiring faculty with the appropriate training.

A multiple correspondence analysis (MCA) of responses was stratified by the Carnegie Classification of the respondent's home institution (Fig 8). As can be seen, faculty at associate's-granting colleges are markedly different from those at the other three institution types in a number of ways. These faculty are the least likely to be including bioinformatics in their

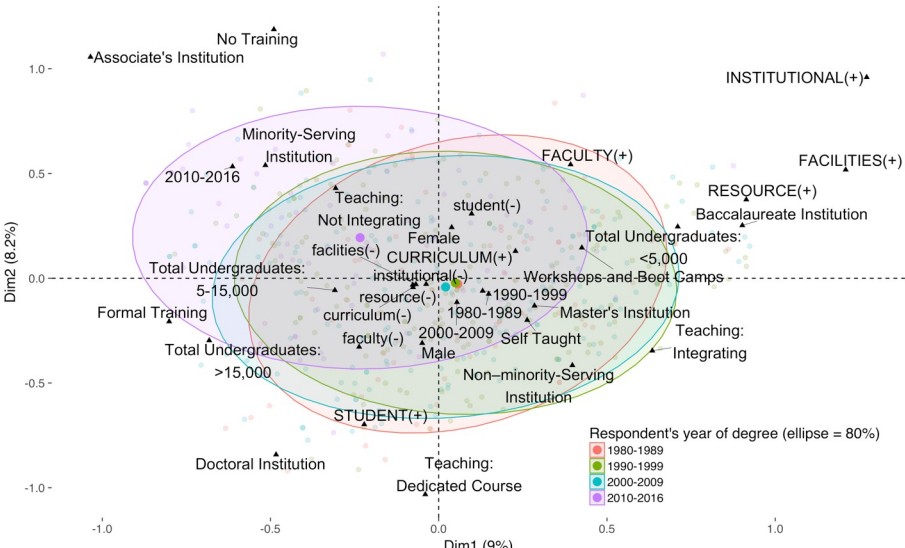

**Fig 5. Multiple correspondence analysis of responses stratified by year of highest degree.** Multiple correspondence analysis allows categorical data to be visualized in a manner similar to the way in which principle component analysis is used for numerical data. Here we display several demographic categories of survey respondents in one figure. A sampling of individual respondents (pale colored dots) are grouped in a colored ellipse encompassing 80% of the respondents in one of four cohorts defined by the decade in which they earned their highest degree (see key); an ellipse is centered on a bold colored dot that represents the average location of all the respondents in that cohort. In the figure, the youngest cohort, terminal degrees earned in 2010–2016, clearly separates from the older cohorts, meaning that the overall experience of this group is different than that of the other three. Only respondents who responded to all the demographic questions are shown ($n = 526$). In addition to information about a respondent's decade of terminal degree, two other types of categorical information are mapped onto the two-dimensional space of the figure. Five demographic categories—1) level of bioinformatics training (No Training, Self-Taught, Workshops and Boot Camps, Formal Training); 2) current bioinformatics content in teaching (Teaching: Dedicated Course, Teaching: Integrating, Teaching: Not Integrating); 3) sex (Female, Male); 4) institution minority-serving status (Minority-serving Institution, Non-Minority-Serving Institution); and 5) undergraduate enrollment (Total Undergraduates < 5,000, Total Undergraduates 5–15,000, Total Undergraduates > 15,000)—are positioned as small black triangles. We also map binary values ("BARRIER (+)," reported the barrier; "barrier (-)," did not report the barrier) for each of the barrier categories reported in free-text Question 1. For example, FACULTY (+) indicates that one of the faculty issues was reported. Holistically, the plot allows correlations between faculty who answered questions in similar ways to be visualized. For example, faculty who earned their terminal degree the most recently (2010–2016) were the least likely to be including bioinformatics in their teaching because ▲Teaching: Not Integrating is near the center of that ellipse and on the edges of the others. Similarly, faculty at minority-serving institutions were more likely to also indicate that they earned their terminal degree in 2010–2016 because ▲Minority-Serving Institution is in the "2010–2016" ellipse and outside of the others. Finally, faculty at doctoral-granting institutions are more likely to indicate they are teaching dedicated bioinformatics courses because ▲Doctoral Institution is closer to ▲Teaching: Dedicated Courses than it is to ▲Teaching: Integrating or ▲Teaching: Not Integrating. Note that black triangle category markings and bold color dots for the same category (e.g., year of degree) are not expected to overlap as this would require a perfect correlation between a single category (e.g., year-of-degree) and all the other mapped categories.

teaching and more likely to report little to no training in bioinformatics, even though bioinformatics skills would contribute to the workforce readiness of their students. In contrast, faculty at doctoral-granting institutions are more likely to have formal training in bioinformatics and to teach dedicated courses in this discipline. They are also the most likely to mention higher-level student issues, such as poor computer science and statistics preparation. Finally, faculty at baccalaureate colleges and master's institutions are more likely to have obtained training via informal modes, such as workshops and boot camps. When a multiple correspondence analysis of responses is stratified by the extent of bioinformatics integration, the three groups are almost completely separated from one another indicating that they are distinctly different (Fig 9).

**Table 5. Placement of faculty with terminal degrees earned in 2010–2016 by institution type.**

| Institution Type | Faculty Respondents (%) |
|---|---|
| Associate's-granting | 15.6 |
| Baccalaureate-granting | 16.1 |
| Master's-granting | 11.2 |
| Doctoral-granting | 11.7 |

Faculty with terminal degrees earned in 2010–2016 shown as the percentage they represent of survey respondents from the given institution type for all decade-of-degree cohorts examined (1980–1989, 1990–1999, 2000–2009, 2010–2016). Faculty in the 2010–2016 cohort are placed nearly equally among the four institution types (no significant difference, $P = 0.289$).

## Discussion

To the best of our knowledge, this is the first study to examine barriers US life sciences faculty face in integrating bioinformatics into undergraduate biology education, and as noted above, it provides a national consensus view on this issue. In our analysis, surveyed faculty over-whelmingly agreed that bioinformatics should be integrated into biology instruction, but only about a third did so. Our work thus provides direct evidence to support the commonly held tenet that a significant majority of life science students earn their degrees without exposure to bioinformatics. Training was reported as the most significant barrier, a finding that held whether the respondent data were stratified by sex, race and ethnicity, Carnegie Classification, MSI- status, the size of the undergraduate population, or geographic region.

We identified several other important trends in our data. First, faculty also often mentioned time as a barrier, although it was clear from the comments in the survey that this meant differ-ent things to different people—time for training, time for instruction (i.e., because there was a great deal of content to cover, it was difficult to find time for instruction on bioinformatics), as

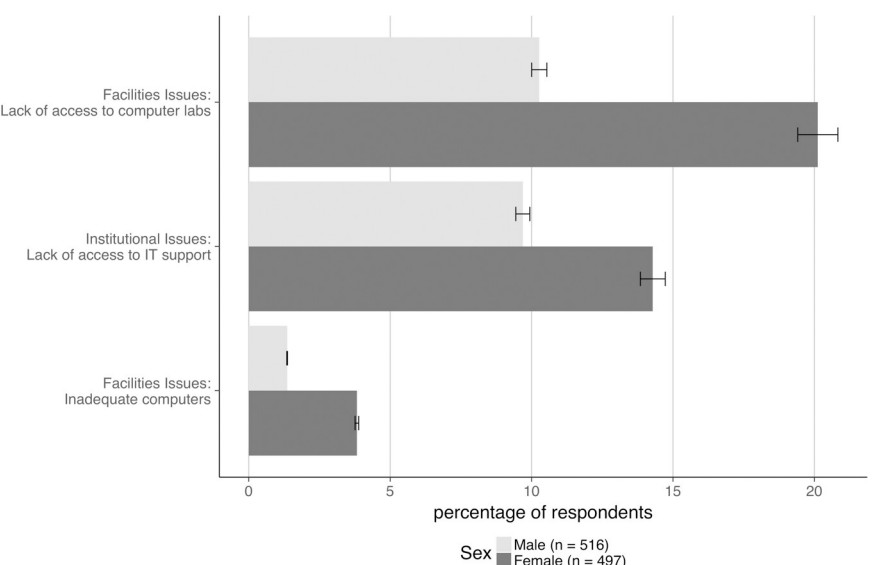

**Fig 6. Barriers reported by females compared to males.** Three barriers to integrating bioinformatics into instruction, all dealing with technology, were reported differently by males and females. As shown in the figure, females reported lack of access to computer labs, lack of information technology (IT) support, and inadequate computer resources at much higher rates than males.

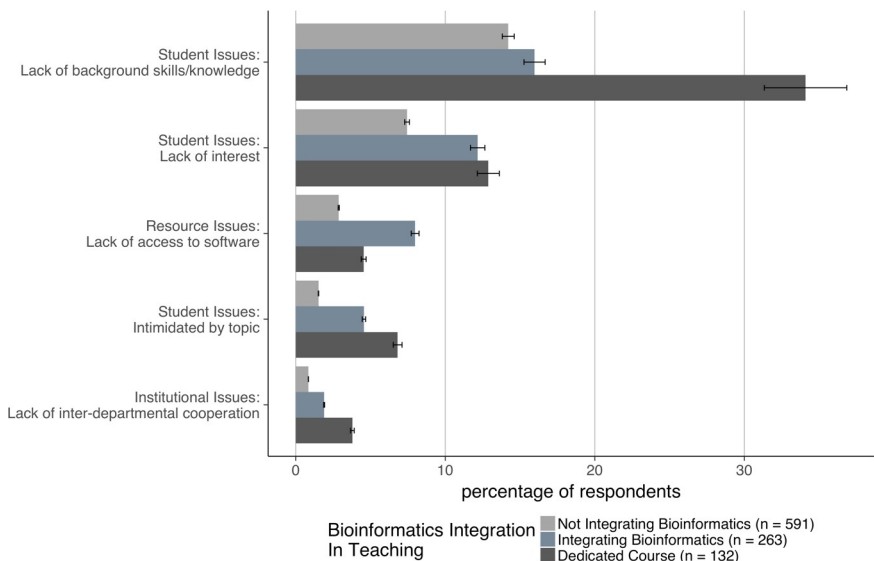

**Fig 7. Types of barriers and extent of bioinformatics integration.** Respondents were asked to indicate how they currently integrate bioinformatics into their teaching if at all ($n = 986$, effect size at 80% power = 0.1, meaning small effects were detected). Of the types of barriers reported by respondents, these five showed significant differences when analyzed by extent of integration (not integrating bioinformatics, integrating bioinformatics, or teaching a dedicated course). Students' lack of background knowledge and skills was most frequently reported as an issue by faculty teaching a dedicated bioinformatics course ($P = 2.7e-7$). Student lack of interest ($P = 0.03$) was reported by a number of faculty. Access to software ($P = 0.003$), student intimidation ($P = 0.001$), and lack of inter-departmental cooperation ($P = 0.03$) were only reported by small numbers of faculty but differed significantly among cohorts.

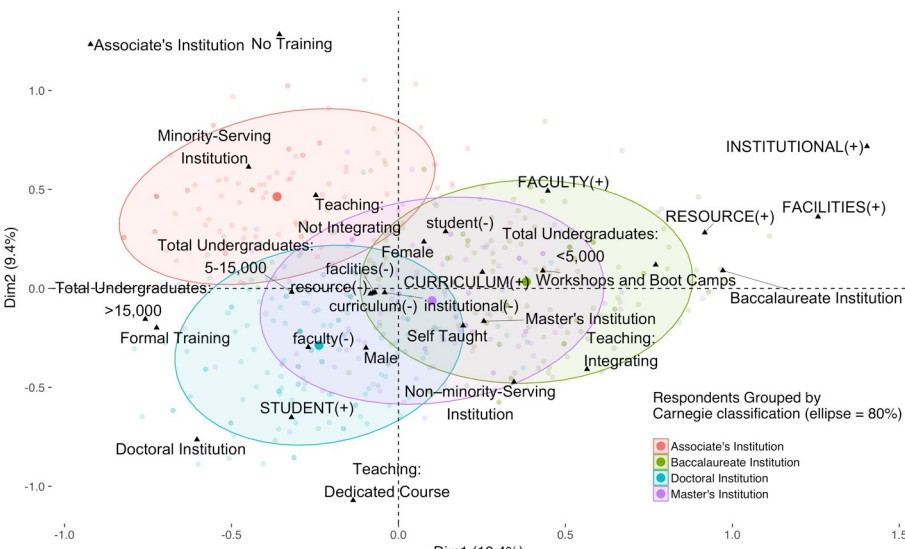

**Fig 8. Multiple correspondence analysis of responses stratified by Carnegie Classification.** Multiple correspondence was calculated grouping faculty by institutional Carnegie Classification (see Fig 5 and Materials and Methods). As mentioned in the narrative, the figure shows that faculty at associate's-granting institutions are different from other institutions in a number of key aspects with respect to barriers to inclusion of bioinformatics in their teaching. In contrast, faculty at the other institution types map along a continuum, with faculty at baccalaureate-granting institutions more likely to integrate bioinformatics into their teaching, faculty at doctoral-granting institutions more likely to teach dedicated bioinformatics courses, and faculty at master's-granting institutions in the middle. Only respondents who responded to all the demographic questions are shown ($n = 526$).

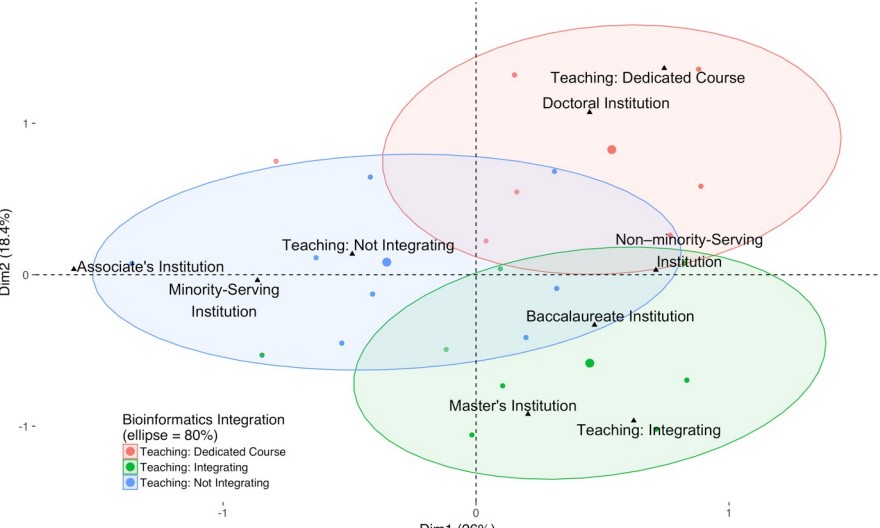

**Fig 9. Multiple correspondence analysis of respondents by integration of bioinformatics, Carnegie Classification, and institutional minority-serving status.** Multiple correspondence was calculated grouping faculty by their level of bioinformatics teaching: teaching a dedicated bioinformatics course (Teaching: Dedicated Course), integrating bioinformatics into existing courses (Teaching: Integrating), and not teaching bioinformatics (Teaching: Not Integrating). (See Fig 5 and Materials and Methods.) Here, the Carnegie Classification of the respondent's institution, illustrated with an upward triangle (▲), was used as the predicted qualitative supplementary factor. The plot reveals that correlations between institution type and the level of bioinformatics teaching separate faculty into three distinct populations. For example, teaching a dedicated course in bioinformatics tends to be associated with doctoral-granting institutions and integrating bioinformatics into existing courses is associated with master's institutions; faculty at associate's colleges tend not to include bioinformatics in their teaching. As discussed in the narrative, faculty at minority-serving institutions face additional barriers in integrating bioinformatics, and as shown in the figure, faculty at these institutions tend not to include bioinformatics in their teaching. Only respondents who responded to all the demographic questions are shown (*n* = 526).

well as time for restructuring the curriculum. We plan to explore these issues further in a future study.

Second, faculty with the most training, the youngest cohort, teach bioinformatics the least. Although faculty at associate's-granting institutions are less likely to integrate bioinformatics in general, we cannot conclude from this that faculty placement is sufficient to explain why the 2010–2016 cohort is the least-likely group to report integrating bioinformatics into their teaching despite better training (Table 5). A potential explanation is that as new faculty they are unable to shape the overall curriculum and/or are not yet tasked with teaching courses that best match their skills. We predict this discrepancy will lessen as this cohort becomes more senior in status and as additional cohorts of PhD trainees become faculty. However, we also note that as long ago as 1998, there were calls for the development of graduate programs in bioinformatics and computational biology [17]. While many such programs at the graduate level have been developed since then [18,19], graduates from these programs appear to have made little impact on biology education at the undergraduate level thus far. It is possible academia is less attractive to individuals fully trained in bioinformatics, who perhaps find better opportunities elsewhere. Preparing faculty that are equally well-trained in the biology, mathematics, computer science, and statistics necessary to teach the breadth of bioinformatics is a long-standing dilemma, although initiatives such as QUBES (Quantitative Undergraduate Biology Education and Synthesis) are making efforts to address this gap [20,21]. However, our findings illustrate more broadly the difficulties inherent in teaching interdisciplinary topics like bioinformatics.

Third, many faculty indicated that students were underprepared to engage in bioinformatics instruction. While faculty at doctoral institutions most often mentioned lack of high-level training in computer science and statistics, faculty at other institutions, especially community colleges, instead cited lack of preparation in basic mathematics skills. Lack of preparedness for college-level mathematics is a longstanding issue for students aspiring to college. In a recent review of the topic, McCormick and Lucas [22] cite a number of studies that describe the scope of the problem. For example, a study from 2001 by Morgan and Michaelides [23] determined that approximately 50% of first-year students were engaged in a remedial mathematics course. These findings suggest that creative ways to include basic mathematics skills in the context of a bioinformatics course are necessary.

Fourth, consistent with percentages of such faculty at institutions around the country [16], our study gathered relatively few respondents (81) who identified as members of groups underrepresented in STEM. Although we are aware that members of individual groups likely have different needs, responses from underrepresented groups were binned together for analysis. Previous reports have noted that at many historically black colleges and universities, bioinformatics courses have not been widely implemented due to a number of factors similar to those outlined here for the wider range of faculty, including lack of faculty training and lack of resources [24]. These trends with regard to faculty at MSIs and URM faculty suggest that serious attention to equity in training opportunities is necessary.

We found a few other trends based on demographics in our data that we need more information to interpret. Faculty at master's institutions were more likely to cite lack of student interest as a barrier (Fig 4). Faculty teaching dedicated courses in bioinformatics more frequently reported that students lack needed background skills and knowledge and are intimidated by the topic. On the other hand, faculty attempting to integrate bioinformatics reported a lack of access to software at higher rates (Fig 7). Some barriers are experienced at higher rates by females than males (Fig 6). We plan to investigate some of these trends in a second study, including the finding that faculty at MSIs experience barriers at a slightly lower rate than non-MSI faculty. In this instance, the difference may be explained by the lower number of faculty at MSIs who are integrating bioinformatics: only 15% of the faculty at MSIs are integrating bioinformatics into their teaching in some way compared to 27% of faculty at non-MSIs ($n = 638$), but we intend to explore this point further.

Other studies have also investigated faculty, student, and institutional barriers to the integration of bioinformatics into life sciences education. Barone, Williams, and Micklos [25], surveying 704 National Science Foundation investigators from the Directorate for Biological Sciences, also found that training was the top unmet need within the research community. Cummings and Temple [19] describe three general categories of challenges for broader incorporation of bioinformatics in education: 1) required infrastructure and logistics; 2) instructor knowledge of bioinformatics and continuing education; and 3) the breadth of bioinformatics and the diversity of students and educational objectives. Barriers we uncovered here with faculty in the United States are also felt by faculty in the United Kingdom [9], as well as in emerging areas more globally [26], specifically in some African countries [10] and in India [11].

What can be done to alleviate barriers? Although a few institutions, such as the University of Wisconsin-La Crosse [27], Kalamazoo College [28], Muhlenberg College [29], and Drake University [30], have reported successful integration of bioinformatics into their life sciences programs [31], the majority of institutions appear not to have done so. Clearly, given that we and others [19,32] have found that lack of faculty training is a major problem, providing faculty with opportunities for training is important, as is giving faculty time to take advantage of these opportunities.

At present, there are many opportunities for faculty training available in the United States and elsewhere. Some of the opportunities include workshops provided by groups such as

BioQUEST (http://bioquest.org); Data Carpentry (http://datacarpentry.org) [33]; DNA Subway (http://dnasubway.cyverse.org); Genome Consortium for Active Teaching (GCAT)-Seek (http://gcat-seek.weebly.com) [34]; Genomics Education Partnership (http://gep.wustl.edu) [35,36]; Genome Solver (http://genomesolver.qubeshub.org) [37]; Integrated Microbial Genomes Annotation Collaboration Toolkit [38,39]; SEA-PHAGES (http://seaphages.org) [40]; Software Carpentry (http://software-carpentry.org); QUBES (http://qubeshub.org); the National Center for Biotechnology Information at the National Institutes of Health (http://ncbi.nlm.nih.gov); the European Bioinformatics Institute (http://www.ebi.ac.uk); the Global Organisation for Bioinformatics Learning, Education, and Training (GOBLET) [9]; and ELIXIR [26]. Such groups are important not only for conveying information and knowledge but for building community. In addition, many schools offer bioinformatics graduate courses and certificates, either in person or online. There are also numerous courses offered in bioinformatics and computer science through Coursera (https://coursera.org) and EdX (https://edx.org). However, finding these training opportunities is left to individual faculty. NIBLSE plans to serve as a clearinghouse for such opportunities. One of our key findings is that faculty who have participated in informal training like workshops or boot camps report the need for training more than faculty with no training or faculty with formal training. This result is similar to that reported by Feldon et al., who suggest that boot camps and short workshops are not very effective for PhD students in the life sciences [41]. It thus may be useful to conduct a follow-up survey to address the deficits expressed by faculty with informal training.

Cummings and Temple [19] recommend "using transformative computer-requiring learning activities, assisting faculty in collecting assessment data on mastery of student learning outcomes, as well as creating more faculty development opportunities that span diverse skill levels, with an emphasis placed on providing resource materials that are kept up-to-date as the field and tools change." NIBLSE is developing a set of teaching tools in its Learning Resource Collection that will help contextualize bioinformatics in light of the fundamentals of biology (http://niblse.org). We also point to the increasing number of resources in the Bioinformatics course on the *CourseSource* website (https://coursesource.org). These two centers of collected resources will also address the concern exhibited by respondents about the difficulty of finding tested curricula to use in their classrooms. We also note that important fundamental concepts in biology, including evolution and the central dogma, could be taught in the context of bioinformatics, helping to alleviate the "too-full curriculum" barrier expressed by some respondents.

To conclude, our results indicate that life sciences faculty overwhelmingly agree that bioinformatics should be integrated into the undergraduate life sciences curriculum, but many barriers exist that prevent them from doing so, a lack of training being the most significant. In addition, our study reveals that the barriers faculty face depend on demographic and other factors. Needs are especially great for members of underrepresented groups in STEM and for faculty at associate's-granting institutions. While many questions about the landscape of bioinformatics education remain, moving forward, NIBLSE seeks to address the challenges uncovered in the present analysis in order to achieve integration of bioinformatics into the life sciences curriculum. The goals articulated by NIBLSE resonate with the recommendations stated in *A New Biology for the 21st Century* to create a community of researchers dedicated to solving a broad range of scientific and societal issues with interdisciplinary approaches and training students to be able to converse across disciplinary boundaries [42].

## Materials and methods

The survey of life sciences faculty was collaboratively developed by a subgroup of NIBLSE members, the Core Competencies Working Group (CCWG). Faculty from a range of

educational institutions were represented in the CCWG, including faculty at baccalaureate-, master's-, and doctoral-granting institutions with various levels of research activity. One of the members of the CCWG was from industry. All members of the working group have extensive experience teaching bioinformatics to undergraduate biology students. Development and deployment of the survey is discussed in more detail by Sayres et al. [12]; the survey in its entirety is provided there as a supplementary document. Approval for the study was obtained from the University of Nebraska at Omaha Institutional Review Board (IRB # 161-16-EX) before the survey was distributed.

The survey was administered in April 2016 using Qualtrics with assistance from the Center for New Designs in Learning and Scholarship at Georgetown University; 1,264 responses were collected. The branched survey design included five-point Likert and free-response questions. As described by Sayres et al. [12], the survey was e-mailed to the more than 11,000 addresses in a mailing list of US biology faculty purchased from MDR (http://schooldata.com) and to members of networks of faculty with interests in life sciences education. Given 75,000 to 100,000 biological sciences faculty in the United States [12] and the total number of responses (1% to 2%), we estimate that the mean margin of error for the survey questions described in this paper is ± 3% at the 95% confidence interval [43]. For the results described here, we analyzed barriers to teaching bioinformatics through four free-response questions (Table 1). The responses were subjected to qualitative analysis by two groups, one at Georgetown University (AGR, using the classic content analysis method outlined in Leech and Onweugbuzie [44]) and one at the University of Florida (JCD, SG, and EWT, using a modification of the coding and thematic analysis process described by Harding [45]). In both analyses, categories of barriers—e.g., "No expertise/training," "Time," "Not enough faculty"—were deductively identified and then combined into super-categories (e.g., "Faculty Issues," "Student Issues," and "Resource Issues") as shown in Table 2 for Question 1. The number of responses that described a given barrier was then counted. Although similar results were obtained from the two analyses, the authors decided to use the data from the University of Florida quantification for detailed analyses because the way in which it was formatted made subsequent analyses easier.

Survey data were exported to CSV-formatted files for analysis in R. Data were cleaned to eliminate multiple column headers and to transform Qualtrics numerical coding of responses into decoded values. During this step, responses from outside the US were eliminated, leaving $n = 1,231$ valid responses. Unless otherwise indicated, we used this number in all calculations. Values smaller than 1,231 occur in two cases: 1) For the four free-response questions, values of $n$ are always the largest number of respondents who could have answered that question (some questions were only asked in particular branches of the survey). Blank responses were conservatively assumed to be intentionally unanswered as it was not possible to tell if a question was simply skipped or if the individual experienced no barriers. 2) Where a statistic involved a multiple-choice question, null responses (i.e., blank, unsure, or "rather not say" responses) were removed from the analysis. In some cases (e.g., respondent race/ethnicity, level of bioinformatics training, and degree year), responses were binned to achieve sufficient numbers for analysis. For example, the responses from respondents who identified as being from a race/ethnic background underrepresented in STEM were analyzed together.

## Analysis methods

The reported barriers were analyzed with respect to a number of demographic criteria—sex, race/ethnicity, highest degree earned, year of highest degree, level of bioinformatics training, extent of current bioinformatics teaching, institutional Carnegie Classification, MSI vs. non-

MSI status, size of school by undergraduate enrollment, and geographic region—to determine differences within these demographics and association of demographics and barriers. For a given demographic, respondents who did not answer, or indicated they did not know or were unsure, were dropped from analysis of that demographic category.

The MCA packages in R were used to visualize the correspondence of several categorical demographic factors [46,47]. Similar to a principle component analysis, MCA allows associations between categorical variables (e.g., our demographic categories) to be visualized. In our analysis, individuals for which we had complete demographic data were used to display relationships in two-dimensional space.

### Proportion tests within demographics

A proportion test was used to calculate the $\chi^2$ statistic for differences between sub-demographics ($H_0$ assuming faculty within all the sub-demographics report barriers equally). The margin of error (as the interval estimate of population proportion) was calculated at the 95% confidence level and is represented on Figs 4, 6 and 7 as error bars. Expected effect sizes detectable were calculated assuming 80% power. Selected findings are described in Results. Additional findings as well as the full data set and R scripts used for analyses and plotting can be found on the NIBLSE GitHub repository available at https://github.com/niblse.

### Supporting information

**S1 Responses. This file contains example responses to the survey questions that probed the barriers life sciences faculty face in integrating bioinformatics.**
(PDF)

### Acknowledgments

The authors thank the members of the Genomics Education Partnership, Genome Solver, GCAT-SEEK, and NIBLSE networks for the feedback they provided. We also thank Drs. Sarah Elgin and Robin Wright for their input in the early stages of this work. AGR thanks Gopal Topiwala for his help with the Georgetown analysis. JCD, SG, and EWT thank Jonathan Orsini for his help with the UF analysis; we also thank Courtney Soderberg and the statistical consulting service at The Center for Open Science.

### Author Contributions

**Formal analysis:** Jason J. Williams, Jennifer C. Drew, Sebastian Galindo-Gonzalez, Eric W. Triplett, Anne G. Rosenwald.

**Funding acquisition:** Elizabeth Dinsdale, William R. Morgan, Eric W. Triplett, Anne G. Rosenwald, Mark A. Pauley.

**Writing – original draft:** Jason J. Williams, Anne G. Rosenwald.

**Writing – review & editing:** Jason J. Williams, Jennifer C. Drew, Sebastian Galindo-Gonzalez, Srebrenka Robic, Elizabeth Dinsdale, William R. Morgan, Eric W. Triplett, James M. Burnette III, Samuel S. Donovan, Edison R. Fowlks, Anya L. Goodman, Nealy F. Grandgenett, Carlos C. Goller, Charles Hauser, John R. Jungck, Jeffrey D. Newman, William R. Pearson, Elizabeth F. Ryder, Michael Sierk, Todd M. Smith, Rafael Tosado-Acevedo, William Tapprich, Tammy C. Tobin, Arlín Toro-Martínez, Lonnie R. Welch, Melissa A. Wilson, David Ebenbach, Mindy McWilliams, Anne G. Rosenwald, Mark A. Pauley.

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
