## [Decision Letter · Decision Letter 0]

21 Aug 2019

PONE-D-19-17542

Barriers to integration of bioinformatics into undergraduate life sciences education: a national study of US life sciences faculty uncover significant barriers to integrating bioinformatics into undergraduate instruction

PLOS ONE

Dear Dr. Pauley,

Thank you for submitting your manuscript to PLOS ONE. After careful consideration, we feel that it has merit but does not fully meet PLOS ONE’s publication criteria as it currently stands. Therefore, we invite you to submit a revised version of the manuscript that addresses the points raised during the review process.

Please read carefully the comments, answer all the issues raised and make changes (as you find appropriate) in the revised version. 

We would appreciate receiving your revised manuscript by Oct 05 2019 11:59PM. To enhance the reproducibility of your results, we recommend that if applicable you deposit your laboratory protocols in protocols.io, where a protocol can be assigned its own identifier (DOI) such that it can be cited independently in the future. For instructions see: http://journals.plos.org/plosone/s/submission-guidelines#loc-laboratory-protocols

We look forward to receiving your revised manuscript.

Kind regards,

Cesario Bianchi

Academic Editor

PLOS ONE

Journal Requirements:

Additional Editor Comments (if provided):

Dear Dr.Pauley:

Thank you for submitting your interesting work. Although both reviewers found the work potentially publishable, I would like to answer all and every issue raises and make the appropriate changes in the revised version.

Thank you

Reviewers' comments:

Reviewer's Responses to Questions

**Comments to the Author**

1. Is the manuscript technically sound, and do the data support the conclusions?

Reviewer #1: Yes

Reviewer #2: Yes

2. Has the statistical analysis been performed appropriately and rigorously? 

Reviewer #1: Yes

Reviewer #2: Yes

3. Have the authors made all data underlying the findings in their manuscript fully available?

Reviewer #1: No

Reviewer #2: Yes

4. Is the manuscript presented in an intelligible fashion and written in standard English?

Reviewer #1: Yes

Reviewer #2: Yes

5. Review Comments to the Author

Reviewer #1: I have some concerns with inconsistencies in the way the methodology is presented in the paper.

In the abstract, the mode of analysis that was used to identify barriers was not mentioned. The writing of the abstract suggests the barriers were identified quantitatively. It is important to make clear in the abstract that this is a mixed methods study that uses qualitative analysis.

It is not clear from the methods description whether the keyword analysis was inductive or deductive i.e. were the keywords and supercategories pre-determined or did they arise from the responses themselves. This is a very important methodological question. Either is fine.

In the introduction (line 136) the analysis of free response questions is described as keyword analysis. In line 177 in results its again described as keyword analysis. Elsewhere in line 175 and 542 its described as qualitative analysis. These are minor distinctions, but since many of the readers will be from the life sciences and have limited knowledge of qualitative approaches - clarity is very important.

Similarly, the methods section has some missing citations:

line 543 citation missing for Leech and Onweugbuzie

line 545 citation missing for Harding

The manuscript also does not include a supplement that shows de-identified example responses that align with each code.

Similarly, the manuscript does not describe any plans for releasing the data, or provide reasons why the data will not be released.

Overall the manuscript and the associated education efforts are an important and laudable contribution to the field.

Reviewer #2: This work addresses an important problem: bioinformatics training. It has done this through a standard survey method. The methodology seems sound to me, and the level of responses obtained seems what one would expect from such a survey. Of course there are limitations when the level of response is 1 or 2% of the population. With that as a caveat, I would say that the results of this survey are likely representative of the intended population.

The main result is that even in 2019 bioinformatics still faces training barriers. This matches my individual experience. Even though in that sense the result is not surprising, taken in perspective (25 years since the H. influenzae genome was published) it *is* surprising. Given the importance of the field, 20 years ago I had the expectation that it would have been much easier to adopt bioinformatics in curricula by now, and this, sadly, is not yet the case. So I think this work will be an important contribution, which will help show university officials that there is a real need to help facilitate the introduction of bioinformatics in curricula.

Minor comments:

1) there are several empty references in Methods. Eg. "...at the 95% confidence interval []".

2) I think the authors should cite (and comment) this report:

A New Biology for the 21st Century

Committee on a New Biology for the 21st Century:

Ensuring the United States Leads the Coming Biology

Revolution; National Research Council, 2009

This is an important document, which makes several specific recommendations. Many of them are related to the development of quantitative skills and interdisciplinary education. Bioinformatics is of course of paramount importance in both. By the way, it is curious that this manuscript does not use the word "interdisciplinary" even once.

3) the definition given for bioinformatics is "a discipline that combines aspects of biology, statistics, and computer

science". Math is left out, and some mathematicians who see themselves as contributing to bioinformatics may take offence. The authors may wish to use a more generic term, such as "exact sciences" or "quantitative sciences" to avoid this problem.

4) I wish the authors would have gone a bit further in their discussion, and have made comments on the more general problem of barriers to any interdisciplinary education. In my view, despite all the rethoric one hears about the importance of interdisciplinary research and education, at the undergraduate classroom level we still suffer from the old discipline divisions (departmental silos certain are a major factor). One would think that this could have been a generational issue, but one of the findings of this work (that recently graduated faculty are the least likely to integrate bioinformatics into their teaching) suggests that it's not; which means that something needs to be done, besides waiting old monodiscipline diehards to retire.

6. PLOS authors have the option to publish the peer review history of their article (what does this mean?). If published, this will include your full peer review and any attached files.

Reviewer #1: No

Reviewer #2: Yes: João Carlos Setubal

---

## [Author Response · Author response to Decision Letter 0]

28 Sep 2019

The manuscript has been revised to address comments from the reviewers. The reviewers' concerns and our responses to them are in the attached rebuttal letter labeled "Response to Reviewers."

---

## [Editor Report · Decision Letter 1]

10 Oct 2019

Barriers to integration of bioinformatics into undergraduate life sciences education: a national study of US life sciences faculty uncover significant barriers to integrating bioinformatics into undergraduate instruction

PONE-D-19-17542R1

Dear Dr. Pauley,

We are pleased to inform you that your manuscript has been judged scientifically suitable for publication and will be formally accepted for publication once it complies with all outstanding technical requirements.

With kind regards,

Cesario Bianchi

Academic Editor

PLOS ONE

Additional Editor Comments (optional):

Dear Dr, Pauley,

Thank you for carefully revising the manuscript according to the reviewers comments. I have recommended acceptance.
---

## [Editor Report · Acceptance letter]

7 Nov 2019

PONE-D-19-17542R1 

Barriers to integration of bioinformatics into undergraduate life sciences education: a national study of US life sciences faculty uncover significant barriers to integrating bioinformatics into undergraduate instruction 

Dear Dr. Pauley:

I am pleased to inform you that your manuscript has been deemed suitable for publication in PLOS ONE. Congratulations! Your manuscript is now with our production department. 

With kind regards,

on behalf of

Dr. Cesario Bianchi 

Academic Editor

PLOS ONE